# Image Processing Techniques for Improving Quality of 3D Profile in Digital Holographic Microscopy Using Deep Learning Algorithm

**DOI:** 10.3390/s24061950

**Published:** 2024-03-19

**Authors:** Hyun-Woo Kim, Myungjin Cho, Min-Chul Lee

**Affiliations:** 1Department of Computer Science and Networks, Kyushu Institute of Technology, 680-4 Kawazu, Iizuka-shi 820-8502, Fukuoka, Japan; kim.hyunwoo547@mail.kyutech.jp; 2School of ICT, Robotics, and Mechanical Engineering, Hankyong National University, Institute of Information and Telecommunication Convergence, 327 Chungang-ro, Anseong 17579, Kyonggi-do, Republic of Korea

**Keywords:** Digital Holographic Microscopy (DHM), Improved Denoising Diffusion Probabilistic Models (IDDPM), noise filtering

## Abstract

Digital Holographic Microscopy (DHM) is a 3D imaging technology widely applied in biology, microelectronics, and medical research. However, the noise generated during the 3D imaging process can affect the accuracy of medical diagnoses. To solve this problem, we proposed several frequency domain filtering algorithms. However, the filtering algorithms we proposed have a limitation in that they can only be applied when the distance between the direct current (DC) spectrum and sidebands are sufficiently far. To address these limitations, among the proposed filtering algorithms, the HiVA algorithm and deep learning algorithm, which effectively filter by distinguishing between noise and detailed information of the object, are used to enable filtering regardless of the distance between the DC spectrum and sidebands. In this paper, a combination of deep learning technology and traditional image processing methods is proposed, aiming to reduce noise in 3D profile imaging using the Improved Denoising Diffusion Probabilistic Models (IDDPM) algorithm.

## 1. Introduction

The development of three-dimensional (3D) imaging technology began with the need for the in-depth observation of complex object shapes and structures. Early 3D imaging relied on physical sectioning and mechanical measurements. These methods were not only time-consuming and labor-intensive, but they also often caused damage to the samples. As technology advanced, non-invasive 3D imaging methods began to emerge, such as computed tomography (CT) and magnetic resonance imaging (MRI) [1,2,3,4,5]. While these methods can provide 3D views of the internal structures of living bodies, they are somewhat limited in terms of resolution and real-time imaging capabilities. The emergence of digital holography (DH) technology has brought revolutionary changes to the field of 3D imaging [1]. Based on the principle of holography, it utilizes lasers or other coherent light sources to record the interference patterns of light reflected or transmitted by an object. The significant advantage of this technology is that it not only records the intensity information of the object but also accurately captures phase information, which is crucial for reconstructing the object’s complete 3D information [6].

Digital holographic microscopy (DHM) is the application of DH in the field of microscopy. Unlike traditional microscopes, DHM offers the capability of 3D imaging without the need for physical sectioning [7,8,9,10,11,12,13,14,15,16,17,18]. It achieves rapid and non-destructive 3D reconstruction of samples through the image processing of holograms. This technology is particularly well-suited for fields such as biomedical research [7], materials science [8], and microelectronics [9]. It enables the observation of details in the microscopic world, including living cells, micro-nano structures, and material surfaces [10]. A key advantage of DHM is its ability to capture complete 3D data in a single exposure. This significantly enhances imaging efficiency and real-time performance. Additionally, DHM allows for post-processing techniques like image enhancement, denoising, and resolution improvement [12,13,14], which further enhance image quality and analytical precision [1].

Obtaining thickness information from a sample using DHM introduces noise from the frequency domain direct current (DC) spectrum and phase errors. Applying wide-window sidebands includes the high spatial frequency components of the sample, but worsens the phase error. Using a narrow window reduces the phase error at the cost of losing high spatial frequency components of the sample. Additionally, alterations in DHM interferometry affect the positions of sidebands, consequently modifying the distance between the DC spectrum and the sidebands [12,13,14]. This situation presents a trade-off between the size of the windowed sideband and the extent of phase error. Traditional filtering methods such as Gaussian and median filtering can reduce this noise [12]. However, traditional filtering algorithms use the height information of pixels surrounding the target pixel. To solve this problem, many researchers have been studied the filtering method of DHM using spatial filtering [19,20], spiral phase filter [21], deep learning techniques [22], etc.

To solve this trade-off, we proposed a high variance pixel average (HiVA) algorithm [13], which outperforms traditional methods in denoising effectiveness. However, this algorithm relies on the windowing of the sidebands and imposes specific requirements on the distance between the sidebands and the DC spectrum, limiting its applicability to holographic images in certain contexts. In this paper, we introduce a novel approach that integrates deep learning algorithms with the HiVA filtering algorithm, aiming to broaden the scope of denoising while preserving its denoising efficacy. Our proposed method utilizes the Improved Denoising Diffusion Probabilistic Models (IDDPM) deep learning algorithm to overcome the noise from the DC spectrum [23,24]. The method proposed in this paper is proceeded with the following steps. First, we obtain 3D profiles from the acquired hologram images. At this time, the interference pattern of the acquired hologram images should be narrowed and adjusted so that the HiVA algorithm can be applied. Next, the other 3D profiles are obtained from the same hologram image using the HiVA algorithm and a training dataset is created. This training dataset is trained on the IDDPM algorithm. Finally, we adjust the experimental setup to obtain hologram image with wide interference patterns and apply it to the IDDPM algorithm to check the filtering results. In addition, we also presents the results of relevant experiments and assessments of image quality in this paper.

This paper is organized as follows. In Section 2, we describe the principles of DHM, and the algorithm of the IDDPM and HiVA. Then, image processing and the experimental setup are described in Section 3. To verify our proposed method, we show the experimental results with a discussion in Section 4. Finally, we conclude with a summary in Section 5.

## 2. Theory

### 2.1. Principle of Digital Holographic Microscopy (DHM)

When a hologram undergoes Fourier transformation, the interference fringes are converted into their frequency spectrum. Interference fringes typically exhibit periodic variations, manifesting as sinusoidal waves in the frequency domain. Furthermore, the Fourier transform solution for the sine function consists of two symmetric delta functions. This can be expressed mathematically as [25]
(1)$$F\left\{\sin\left(k_{0}x\right)\right\}=\pi \left[\delta \left(k+k_{0}\right)-\delta \left(k-k_{0}\right)\right]$$
where F{} is the Fourier transform operator. The resulting two peaks obtained are called sidebands. Besides, in the recorded hologram, an object wave O(x,y) and reference wave R(x,y) with complex amplitudes can be represented as [15]
(2)$$O\left(x,y\right)=A_{o}\left(x,y\right)e^{-j\phi_{o}\left(x,y\right)},$$
(3)$$R\left(x,y\right)=A_{r}\left(x,y\right)e^{-j\phi_{r}\left(x,y\right)},$$
where $A_{o}\left(x,y\right)$, $A_{r}\left(x,y\right)$, and $\phi_{o}$, $\phi_{r}$ are the amplitude and phase of the object and reference waves, respectively. On the image sensor, the object wave and the reference wave are coherently superimposed to form a recorded hologram $I_{holo}\left(x,y\right)$. This is expressed as $I_{holo}\left(x,y\right)={\left|R\left(x,y\right)+O\left(x,y\right)\right|}^{2}$. This can be expanded as [15]
(4)$$I_{holo}\left(x,y\right)={\left|R\left(x,y\right)\right|}^{2}+{\left|O\left(x,y\right)\right|}^{2}+R^{*}Oe^{j\phi }+RO^{*}e^{-j\phi }$$
where ∗ represents the complex conjugate, and ϕ is the phase information of the recorded hologram image. Furthermore, the ${\left|R\left(x,y\right)\right|}^{2}+{\left|O\left(x,y\right)\right|}^{2}$ term is the DC spectrum, and the $R^{*}Oe^{j\phi }+RO^{*}e^{-j\phi }$ term represents the positive and negative sidebands of the Fourier domain, respectively. Here, when an object is located in the object wave path of the recorded hologram image, it is called an object image, and when there is no object, it is called a reference image. The phase information of the recorded hologram is obtained from the sidebands. Typically, the extraction of phase information is achieved by applying windowing followed by an inverse Fourier transform, as expressed by Equation (Equation 4). Therefore, phase difference information can be obtained from the difference in phase information between the reference image and object image, and can be expressed as follows:(5)$$\Delta \phi \left(x,y\right)=\phi_{O}\left(x,y\right)-\phi_{R}\left(x,y\right)$$
where $\phi_{O}\left(x,y\right)$ and $\phi_{R}\left(x,y\right)$ are the phase information of the object and reference image, respectively, and Δ is the difference symbol. Therefore, Δϕ is the phase difference between the phase information of the object image and reference image.

By using this phase difference, the height information of the specimen can be obtained, making 3D reconstruction possible. The height information of the specimen, denoted as *h*, can be derived as the following [12,13,14]
(6)$$h\left(x,y\right)=\frac{\Delta \phi (x,y)}{K\Delta n}$$
where *K* is the wavenumber, Δn is the constant refractive index difference between the object and the surrounding medium.

A wavefront (object image) deformed by variations in the refractive index is captured by the image sensor as shown in Figure 1. By comparing it with the wavefront that has not been deformed (reference image), the 3D information of the object can be obtained [12,13,14]. The 3D information of the hologram is further refined by applying the magnification data of the objective lens for the micro-object to the acquired height information.

### 2.2. Image Processing of DHM

Figure 2 depicts a cell object image recorded by using a Mach–Zehnder interferometer alongside a reference image without a sample. The target sample intended for imaging is cropped at the same position in both images, followed by Fourier transform to obtain the frequency domain image as shown in Figure 3a.

The phase information of the hologram is obtained by the inverse Fourier transform of the windowed sidebands. However, as the size of the windowed sidebands is reduced, the resolution of the obtained phase information image also decreases. This is because the resolution of the windowed sidebands is directly proportional to the resolution of the phase information image after inverse Fourier transform as shown in Figure 3. In order to avoid phase errors affected by the distribution of the DC spectrum and the position of the sidebands, the size of the windowed sidebands must be minimized. This scenario raises the issue that the resolution of the resulting phase information image is too small. To address this, a zero matrix with the same resolution as the original hologram is needed as shown in Figure 3b. Then, by placing the windowed sidebands at the center of this matrix and performing an inverse Fourier transform, phase information with the same resolution as the original hologram is obtained. This image processing technique is called as zero padding or Fourier shift [12].

Inverse Fourier transform is performed on the windowed sideband. This step reverts the hologram from the frequency domain back to the spatial domain, but now it encompasses more refined phase information. The result of inverse Fourier transform is a complex image, with its real and imaginary spectrum representing the amplitude and phase information of the sample, respectively. By extracting the imaginary part of the image post inverse Fourier transform, the phase information map of the sample can be obtained. This phase information map reveals the sample’s 3D morphology and variations in optical thickness as shown in Figure 4.

Finally, the phase difference between the object image and the reference image can be calculated as depicted in the Figure 5.

As shown in Figure 5, phase wrapping is observed. This occurs because phase values are typically derived by calculating the arctangent, limiting phase range to −π to π. When the phase shifts beyond this range, it wraps back into this interval, leading to a loss of information. In numerous practical applications, particularly when the accurate measurement of an object’s height profile is required, the absolute value of the phase is essential. Consequently, a method to unwrap these phase values is necessary. In this paper, we used a Goldstein phase unwrapping algorithm [26] to obtain the phase information of the object. After phase unwrapping, the 3D profile of the sample can be obtained as shown in Figure 6. However, the 3D profile in Figure 6 contains noise, which may interfere with diagnosing the disease. From the next subsection, we describes algorithms for solving this problem.

### 2.3. Improved Denoising Diffusion Probabilistic Models (IDDPM)

IDDPM is a generative model used for data denoising and generation. The denoising nature of the model relies on a probabilistic framework that combines diffusion and denoising processes. Its core idea is to gradually transform the data from their original distribution into a simple and known distribution (usually a Gaussian distribution) through a random diffusion process, and then reconstruct the original data through a reverse diffusion process, as shown in Figure 7 [23,24].

The diffusion process, which elucidates the stepwise transformation of data from its original distribution to a Gaussian noise distribution, typically incorporates conditional probability distributions. Given a data distribution $x_{0}∼q\left(x_{0}\right)$, we define a forward noise process *q* that adds Gaussian noise at time *t* with variance $\beta_{t}\in \left(0,1\right)$ to generate potential $x_{1}$ through $x_{T}$ as follows:(7)$$q\left(x_{1},\ldots ,x_{T}|x_{0}\right)=\prod_{t=1}^{T}q\left(x_{t}|x_{t-1}\right)$$
(8)$$q\left(x_{t}|x_{t-1}\right)=N\left(x_{t};\sqrt{1-\beta_{t}}x_{t-1},\beta_{t}I\right)$$
Given sufficiently large *T* and a suitable schedule of $beta_{t}$, the potential $x_{T}$ is nearly an isotropic Gaussian distribution. Therefore, if we know the exact inverse distribution (about the inverse process of reconstructing the original data from noisy data) $q(x_{t-1}|x_{t})$, we can sample $x_{T}∼N\left(0,I\right)$ and run the process in reverse to take a sample from $q(x_{0})$. However, since $q(x_{t-1}|x_{t})$ depends on the distribution over the data, the neural network used is approximately as follows:(9)$$p_{\theta }\left(x_{t-1}|x_{t}\right)=N\left(x_{t-1};\mu_{\theta }\left(x_{t}\right),\Sigma_{\theta }\left(x_{t}\right)\right)$$
The combination of *q* and *p* is a variational autoencoder [23], which is the original noise formula, and the variational lower bound (VLB) can be expressed as follows: (10)$$\begin{array}{cc} L_{\mathrm{vb}} & =L_{0}+L_{1}+\ldots +L_{T-1}+L_{T} \end{array}$$(11)$$\begin{array}{cc} L_{0} & =-\log p_{\theta }\left(x_{0}|x_{1}\right) \end{array}$$(12)$$\begin{array}{cc} L_{T-1} & =D_{KL}(q\left(z_{T-1}|x_{T-1},x_{0}\right)\left|\right|p_{\theta }\left(z_{T-1}|x_{T}\right)) \end{array}$$(13)$$\begin{array}{cc} L_{T} & =D_{KL}(q\left(z_{T}|x_{0}\right)||p\left(z_{T}\right)) \end{array}$$

Except for $L_{0}$, each term in Equation (Equation 10) is the KL (Kullback–Leibler) divergence between two Gaussian distributions, so it can thus be evaluated in closed form, used to measure the model performance. As pointed out in [23], the noise processing defined in Equation (Equation 8) allows us to sample an arbitrary step of the noised patients directly conditioned on the input $x_{0}$. With $\alpha_{t}:=1-\beta_{t}$ and , we can drive the marginal distribution as follow:(14)$$q\left(x_{t}|x_{0}\right)=N\left(x_{t};\sqrt{{\bar{\alpha }}_{t}}x_{0},\left(1-{\bar{\alpha }}_{t}\right)I\right)$$
Here, $1-{\bar{\alpha }}_{t}$ indicates us the variance of the noise for an arbitrary time step. This can be used equivalently to define a noise schedule instead of $\beta_{t}$. The network can predict $x_{0}$, and this output can then be fed through Equation (Equation 10) to produce $\mu_{\theta }\left(x_{t},t\right)$. The network can also predict the noise ϵ added to $x_{0}$, and this noise can be used to predict $x_{0}$ in the following way:(15)$$x_{0}=\frac{1}{\sqrt{\alpha_{t}}}\left(x_{t}-\frac{\beta_{t}}{\sqrt{1-\alpha_{t}}}\epsilon \right)$$

Ho et al. (2020) [23] found that ϵ prediction performs best, especially when combined with a reweighted loss function:(16)$$L_{\mathrm{simple}}=E_{x_{0},\epsilon }\left[{∥\epsilon -\epsilon_{\theta }\left(x_{0},t\right)∥}^{2}\right]$$

In conclusion, within this model, the predictor of $x_{0}$ noise is the reconstruction loss. This loss function evaluates the model’s performance in the denoising procedure, specifically its ability to reconstruct the original data from noisy data. This loss function is defined as the difference between the reconstructed image and the original image, utilizing the mean square error (MSE). To ensure that the model’s stochastic generation process can produce reasonable noise, the KL divergence loss function is incorporated. Consequently, the loss functions employed in this model consist of $L_{KL}$ + $L_{MSE}$.

### 2.4. High-Variance Pixel Averaging (HiVA)

In the context of 3D imaging performed by DHM, during the reconstruction of the 3D phase information of the sample, a trade-off arises. When a narrow area of the sideband is windowed in the frequency domain, it leads to a reduction in the phase error of the DC spectrum but results in the loss of high spatial frequency portions. This leads to the loss of high-frequency details in the sample. If a wide area is windowed, the 3D profile includes high spatial frequency components, but at the expense of increased phase error. To address this challenge, the HiVA method was devised. This approach leverages the variance map of the reconstructed depth profile in the frequency domain, utilizing window sidebands of varying sizes to distinguish between phase errors and high spatial frequency components. This methodology enables the retrieval of high-frequency detailed information from the sample while simultaneously mitigating noise stemming from phase errors. The following provides a description of the HiVA principle [13].

The HiVA approach involves segmenting the 2D spatial frequency domain of both the reference image and object image, transitioning from a narrower region to a wider region at regular intervals. A variance map is generated using phase reconstruction. The average of high-variance pixels is computed. This is because high-variance pixels contain noise originating from the DC spectrum. In addition, for the nonaveraged pixels, the reconstructed phase data generated by the spatial frequency components of the widest window is used. The HiVA formulation is expressed as follows [13]:(17)$$S\left(x,y\right)=\frac{1}{N_{d}}\times \sum_{i=1}^{N_{d}}\frac{\Delta \phi_{i}\left(x,y\right)}{K\times \Delta n}$$
(18)$$V\left(x,y\right)=\frac{1}{N_{d}}\times \sum_{i=1}^{N_{d}}{\left(\frac{\Delta \phi_{i}\left(x,y\right)}{K\times \Delta n}-S\left(x,y\right)\right)}^{2}$$
(19)$$h_{HiVA}\left(x,y\right)=\left\{\begin{array}{cc} \frac{\Delta \phi_{N_{d}}\left(x,y\right)}{K\times \Delta n} & V\left(x,y\right)<m_{V},\\ \frac{1}{N_{d}}\times \sum_{i=1}^{N_{d}}\frac{\Delta \phi_{i}\left(x,y\right)}{K\times \Delta n} & V\left(x,y\right)\geq m_{V}, \end{array}\right.$$
where *S* and *V* are the mean and variance, respectively. Moreover, $h_{HiVA}\left(x,y\right)$ is the height information of the specimen, $m_{V}$ is the average of the variance, $N_{d}$ is the number of phase reconstruction data for windowed sidebands of different sizes. The average of the variance $m_{V}$ is used as the threshold to segment the high-variance pixels and the other pixels in Equation (Equation 19).
(20)$$h_{HiVA}\left(x,y\right)=\left\{\begin{array}{cc} \frac{\Delta \phi_{N_{d}}\left(x,y\right)}{K\times \Delta n} & logV\left(x,y\right)<m_{LV},\\ \frac{1}{N_{d}}\times \sum_{i=1}^{N_{d}}\frac{\Delta \phi_{i}\left(x,y\right)}{K\times \Delta n} & logV\left(x,y\right)\geq m_{LV}, \end{array}\right.$$

However, due to the substantial disparity between the maximum value and other values in the variance plot, conventional thresholding methods cannot be applied for segmentation. A new threshold for segmentation is established based on the mean of the logarithmic variance ($m_{LV}$), as depicted in the Equation (Equation 20). Pixels below this threshold are substituted with zeros to eliminate noise.

### 2.5. Problem of the HiVA and the Proposed Method

HiVA windows the spatial frequency domain from the narrow region to the wide one at regular intervals and then creates a variance map using phase reconstruction. Therefore, HiVA can be used for filtering and denoising only when there is sufficient distance between the DC spectrum in the hologram and the sidebands on both sides, as illustrated below.

In Figure 8a, it is evident that the distance between the DC spectrum and the sideband is sufficient to compute the HiVA variance diagram. Figure 8b lacks the necessary distance to calculate the variance map. Hence, a proposition has been advanced to amalgamate the deep learning IDDPM algorithm with HiVA. This integration aims to extend the high-quality denoising capabilities of HiVA to the scenario illustrated in Figure 8b. The conceptual diagram of our proposal is presented below.

In Figure 9, we describe the model principles and the proposed methodology. The noise data serves as the conditional input, while non-noise data (representing the target data after denoising using HiVA) is utilized for loss function computation. This entails incorporating both noisy and clean data throughout the model training process. To elaborate, during the model training phase, the noise data is employed as the model’s conditional input. A noise image is generated, and a comparison is made between the generated noise image and its corresponding non-noise counterpart to calculate the loss. This process facilitates the model’s learning to effectively eliminate noise, thereby ensuring that the generated image closely resembles the noise-free target image. The following section will validate the proposed method and conduct an assessment of image quality.

## 3. Experimental Setup

Figure 10 illustrates the experimental setup for this section. The dataset used in the model training process of this paper is a hologram image recorded using the experimental setup shown in Figure 10. The modified Mach–Zehnder interferometer using two spherical waves is utilized in this paper. We adjusted the spacing of the fringe pattern by changing the angle of $BS_{2}$ to distinguish between zero order and side orders in the Fourier domain and to create an environment in which it is difficult to apply the HiVA algorithm. In Figure 10, *L* represents the lens, *P* denotes the pinhole, *M* stands for the mirror, and OL and BS correspond to the objective lens and the beam splitter, respectively. The setup includes a Mach–Zehnder interferometer equipped with a spatial filter positioned in front of a laser. A 532 nm green semiconductor laser was utilized. The laser, after passing through the spatial filter, traversed a collimating lens (L2) to produce a parallel wavefront. The final diameter of the laser beam through L2 was 2 mm, with an output power of approximately 3 mW. The two objective lenses employed in this setup were 40 × (0.65 NA) with a working distance of 0.6 mm. Additionally, the exposure time was set to 35 μs. For the experiment, 10 μm polystyrene microspheres (02706–AB, SPI Supplies, West Chester, PA, USA) were placed and spread on a slideglass. To record the hologram, we use a CMOS sensor (acA2500–14uc, Basler, Ahrensburg, Germany) with a pixel resolution of 2588(H) × 1940(V).

In this experiment, an initial noisy dataset devoid of any filtering or denoising is employed as the input. Noise images are generated through the utilization of a pretrained IDDPM model using PyTorch, and the loss function is computed during the generation of these noise images. This process involves the incorporation of the target image, which represents a clean 3D profile denoised using HiVA. The objective is to fine-tune the model, facilitating its acquisition of the correct data reconstruction and denoising procedures.

Following the Table 1, the 3D profile, which has not undergone filtering or denoising, is utilized as input, and the 3D profile is generated using the frequency domain representation as depicted in Figure 8b. During the dataset creation process, adjustments are made to the spacing of interference fringes to regulate the distance from the DC spectrum to the sideband. In the generation of a 3D profile, varying sideband window sizes are also adapted to acquire diverse types of noise and profile images featuring different frequency domain information. Additionally, image rotation is applied as part of the preprocessing techniques. These methods collectively aim to enhance the diversity of the training data and serve as data augmentation measures, effectively guarding against overfitting.

## 4. Experimental Result

Figure 11 presented below provides a comparison of denoising outcomes achieved through three different methods: without filtering, employing HiVA filtering, and utilizing the conventional Gaussian filtering approach. The reason why Gaussian filtering was selected as the comparison target is that the efficiency of Gaussian filtering was the highest in the results when the 3D profile of DHM was filtered using Gaussian, Wiener, average, median, and bilateral filters in [12]. Moreover, in [12,13], it can be seen that the HiVA algorithm shows higher filtering efficiency than Gaussian filtering. It does not seem to be much difference in the filtering results shown in Figure 11b,c. However, we can recognize that HiVA result removes noise better in the background of the enlarged part. As evident from Figure 11b,c, it is discernible that HiVA algorithm yields superior denoising outcomes compared to Gaussian filtering. The higher the Sigma value of Gaussian filtering, the higher the filtering effect will be, but since high frequency information is lost, a similar Sigma value was applied. For the training phase, the HiVA-filtered image in Figure 11c is utilized. However, during testing, images that cannot be effectively filtered by HiVA are employed, as exemplified in Figure 8b.

The training results are depicted in the Figure 12 presented below. As observed in the magnified section of the Figure 12c, the proposed method effectively mitigates noise. Furthermore, within the circled area, it becomes evident that the filtering result achieved by the proposed method as shown in Figure 12c outperforms Gaussian filtering in terms of denoising efficacy while preserving the original profile information of the sample.

Table 2 represent numerical results for the unfiltered image, the Gaussian-filtered image, and the image by our proposed method, respectively. These results are compared to the ideal microsphere image created which height and width are equal to the height of the microsphere [14], and the results of quality assessment in terms of peak signal-to-noise ratio (PSNR) and structural similarity (SSIM) are presented [27,28]. Figure 13 and Figure 14 show the graph schematizing Table 2. In Figure 13 and Figure 14, each horizontal line shows the average value of each evaluation value. It is evident that the proposed method surpasses the conventional Gaussian filter algorithm and exhibits a closer resemblance to the ideal microsphere image.

## 5. Conclusions

In this paper, the 3D profile quality of DHM imaging has been improved using a deep learning algorithm. The improvement involves the integration of the IDDPM algorithm with the HiVA filtering algorithm. This integration expands the applicability of HiVA filtering while retaining its denoising capabilities, surpassing traditional algorithms in denoising effectiveness. To assess this approach, image quality evaluations have been conducted using microspheres.

HiVA demonstrates superior denoising capabilities compared to traditional filtering methods, but its applicability is limited. To overcome this limitation, the IDDPM denoising algorithm has been introduced in conjunction with HiVA, extending the range of its application while preserving HiVA’s denoising proficiency. Evaluations have been conducted using microsphere samples that have been resistant to denoising with HiVA, and the results have been assessed for quality. The proposed method yielded higher SSIM and PSNR values compared to traditional filtering algorithms. As a result, we can say that our proposed method can be used even when the sidebands are close to the DC spectrum. This can contribute to improved precision in medical disease diagnosis. Furthermore, it has been envisaged that holographic digital microscopes can find broader applications across various fields, including biomedicine, materials science, microelectronics, and other high-precision imaging domains.

In terms of future considerations, certain challenges need to be addressed. Due to the intricacy involved in creating cell datasets, their utilization for training purposes is currently unfeasible. The existing 3D profile denoised images lack comprehensive 3D observations from all angles, necessitating the acquisition of more extensive and meticulous datasets for training. Furthermore, the application of advanced image segmentation algorithms, such as the segment anything model (SAM) [29], is imperative for processing a substantial volume of cell datasets to obtain either 3D or 2D images. These challenges lie ahead in our research endeavors.

## Figures and Tables

**Figure 1 sensors-24-01950-f001:**
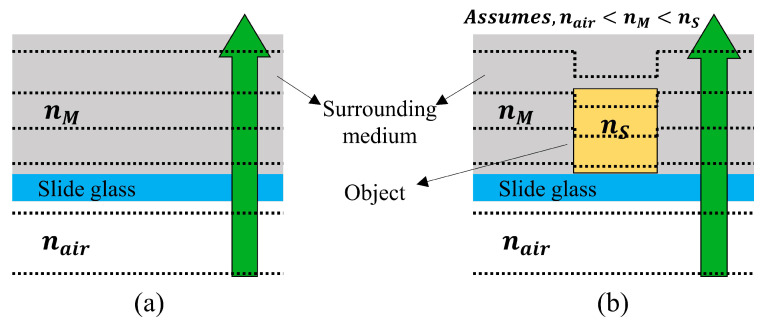
Wavefront scattering in the specimen. (**a**) the wavefront of the reference image and (**b**) the wavefront of the object image, where $n_{a}$: refractive index of air, $n_{M}$: refractive index of the surrounding medium, $n_{S}$: refractive index of the object.

**Figure 2 sensors-24-01950-f002:**
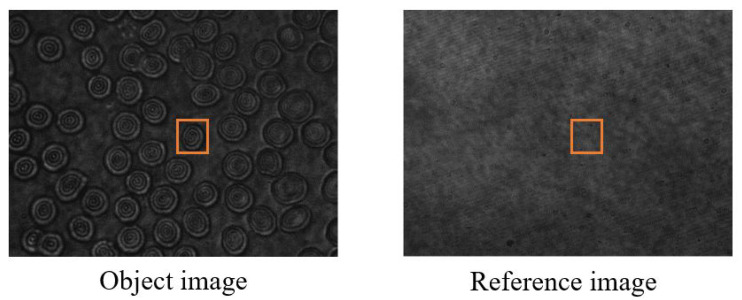
Acquired object and reference holograms of the red blood cells (RBCs).

**Figure 3 sensors-24-01950-f003:**
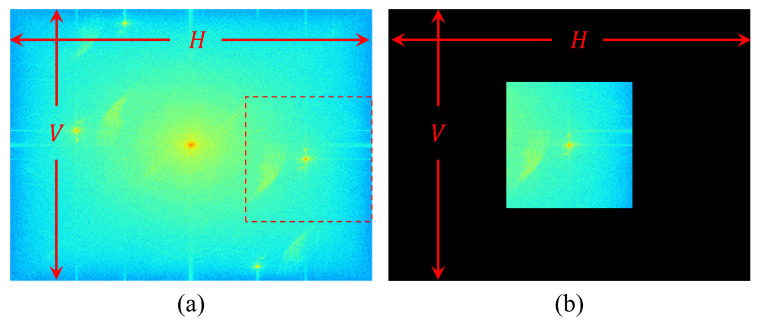
Fourier shift process of DHM. (**a**) Fourier domain of the recorded hologram and (**b**) the windowed sideband from (**a**), where H: horizontal resolution, V: vertical resolution.

**Figure 4 sensors-24-01950-f004:**
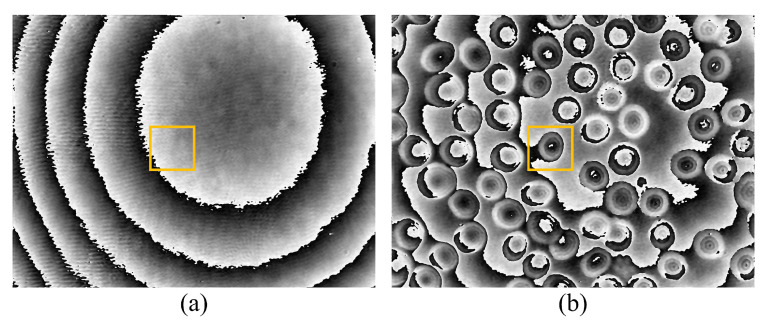
Phase information of (**a**) the reference image and (**b**) the object image after inverse Fourier transform.

**Figure 5 sensors-24-01950-f005:**
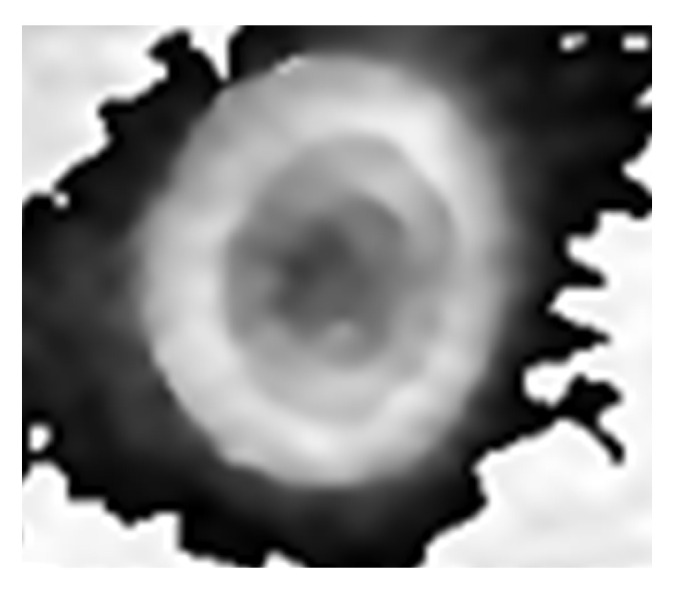
Phase difference.

**Figure 6 sensors-24-01950-f006:**
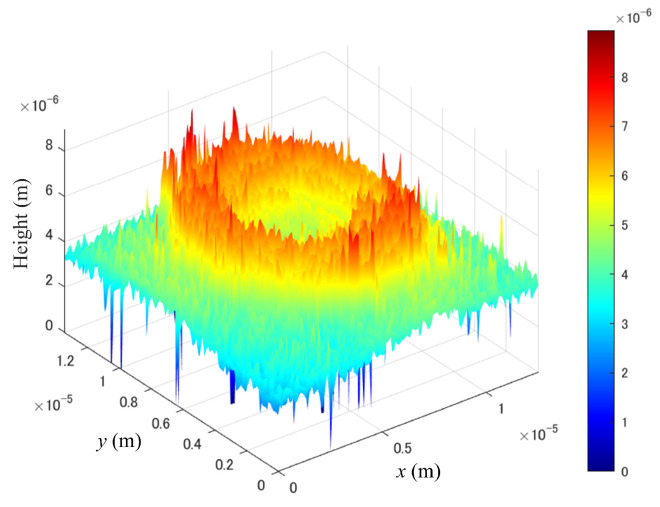
3D profile of the red blood cell.

**Figure 7 sensors-24-01950-f007:**
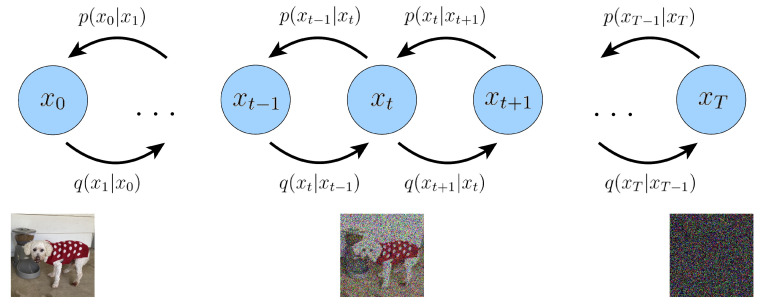
Concept of the Diffusion Model.

**Figure 8 sensors-24-01950-f008:**
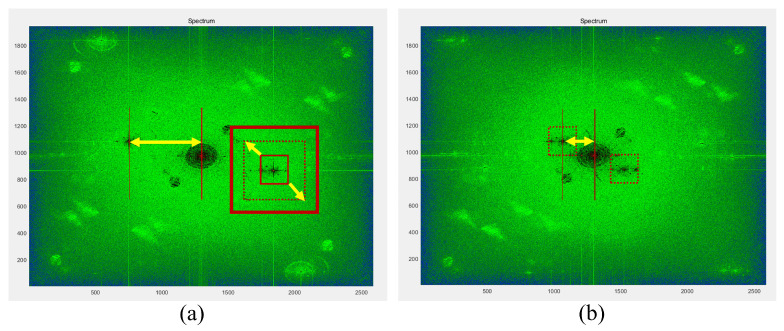
Frequency domain that (**a**) can be filtered with HiVA (**b**) cannot be filtered with HiVA.

**Figure 9 sensors-24-01950-f009:**
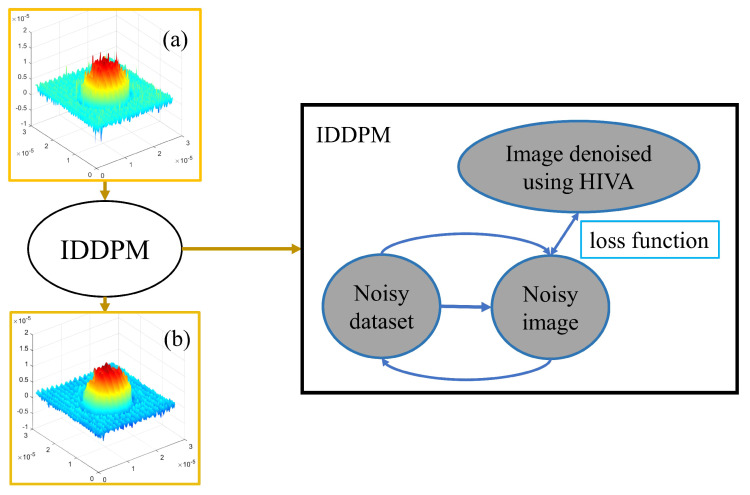
Concept of the proposed method. (**a**) The original 3D profile and (**b**) the filtered 3D profile by HiVA algorithm.

**Figure 10 sensors-24-01950-f010:**
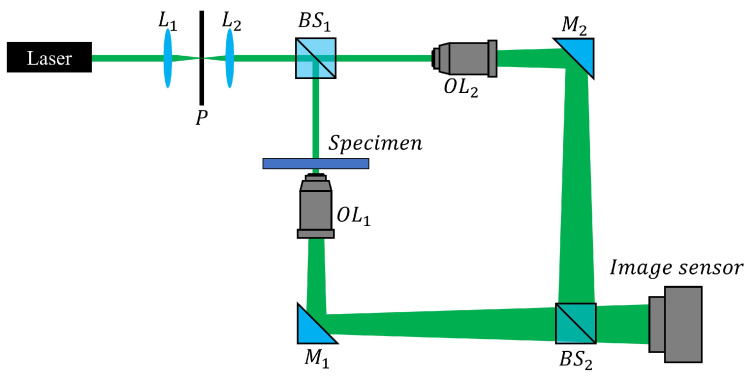
Experimental setup. (*L* : Lens, *P*: pinhole, *M*: mirror, BS: beam splitter, and OL: objective lens).

**Figure 11 sensors-24-01950-f011:**
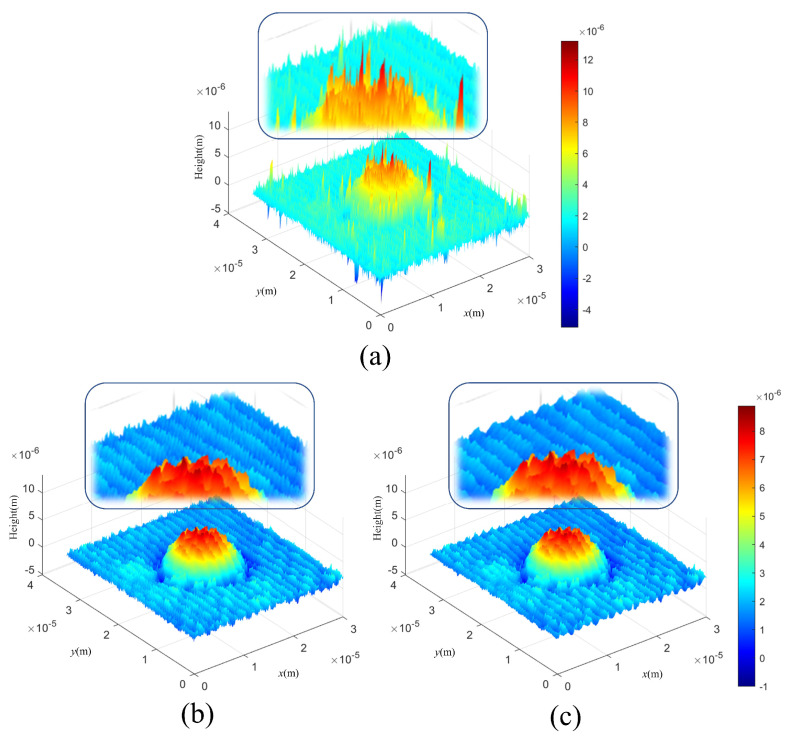
Comparison of filtering results. (**a**) Unfiltered image, (**b**) Gaussian filtering (σ=2), and (**c**) HiVA filtering.

**Figure 12 sensors-24-01950-f012:**
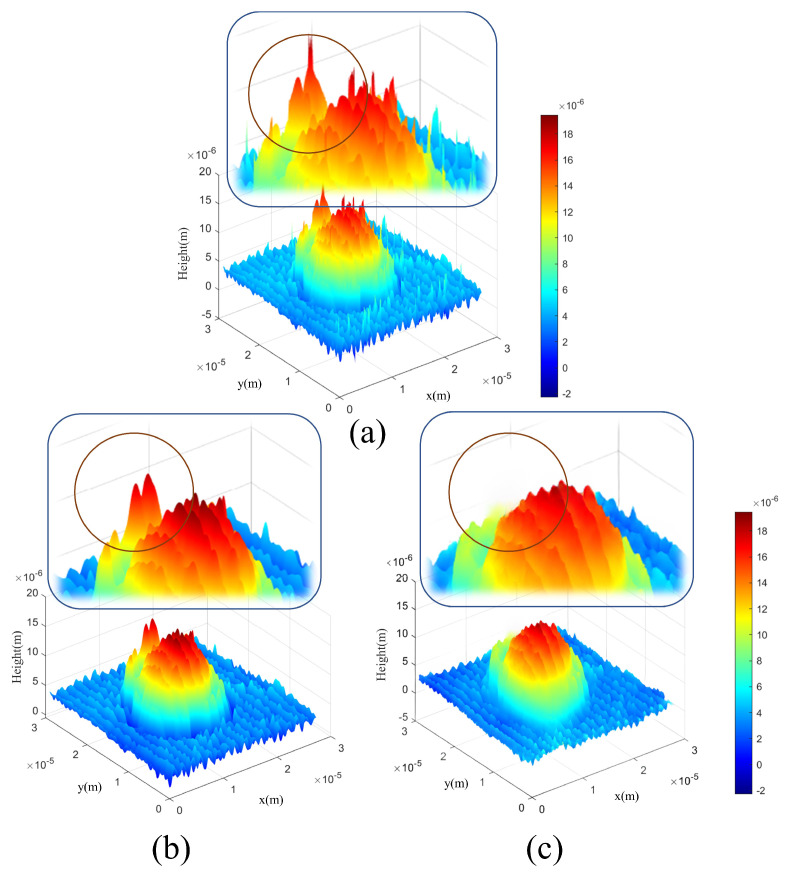
Reconstructed microsphere 3D profile of (**a**) the unfiltered image, (**b**) images filtered with the Gaussian method and (**c**) images filtered with the proposed method.

**Figure 13 sensors-24-01950-f013:**
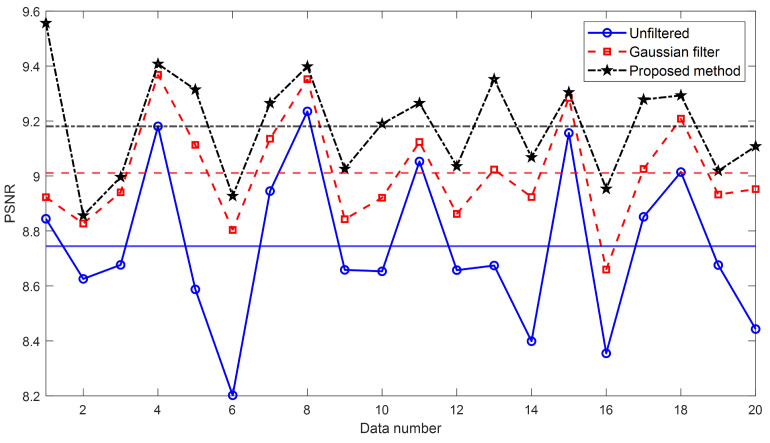
Peak signal-to-noise ratio (PSNR) value of the reconstructed 3D profiles of the unfiltered image, the Gaussian-filtered image, and the image filtered by our proposed method.

**Figure 14 sensors-24-01950-f014:**
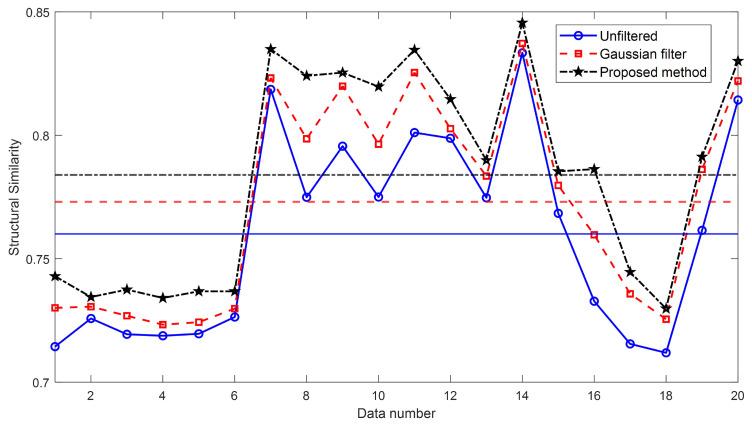
Structural similarity (SSIM) value of the reconstructed 3D profiles of the unfiltered image, the Gaussian-filtered image, and the image filtered by our proposed method.

**Table 1 sensors-24-01950-t001:** Training information.

Hyperparameter	Value
Number of image	2000
Image resolution	875 (H) × 656 (V)
Optimizer	Adam
Number of iterations	80,000
Number of batch size	8
GPU	RTX 3090 12 GB

**Table 2 sensors-24-01950-t002:** Numerical comparison by calculating SSIM and PSNR.

	SSIM			PSNR		
Data Number	Unfiltered	Gaussian Filter	Proposed Method	Unfiltered	Gaussian Filter	Proposed Method
1	0.7144	0.7301	0.7429	8.844	8.923	9.556
2	0.7258	0.7306	0.7345	8.626	8.827	8.856
3	0.7194	0.7269	0.7375	8.677	8.941	8.996
4	0.7188	0.7233	0.7341	9.181	9.369	9.408
5	0.7196	0.7243	0.7368	8.587	9.113	9.314
6	0.7264	0.7298	0.7368	8.202	8.804	8.927
7	0.8186	0.8232	0.8349	8.945	9.135	9.265
8	0.7749	0.7986	0.8241	9.235	9.352	9.399
9	0.7956	0.8199	0.8254	8.658	8.843	9.026
10	0.7750	0.7964	0.8197	8.653	8.921	9.190
11	0.8011	0.8254	0.8346	9.053	9.124	9.265
12	0.7988	0.8027	0.8146	8.657	8.862	9.035
13	0.7747	0.7835	0.7899	8.674	9.024	9.352
14	0.8334	0.8372	0.8456	8.399	8.923	9.068
15	0.7684	0.7797	0.7854	9.156	9.284	9.305
16	0.7328	0.7597	0.7863	8.355	8.659	8.953
17	0.7155	0.7358	0.7446	8.851	9.025	9.278
18	0.7119	0.7255	0.7298	9.014	9.208	9.293
19	0.7615	0.7862	0.7913	8.676	8.933	9.019
20	0.8143	0.8220	0.8301	8.443	8.952	9.108
Average	0.7600	0.7730	0.7839	8.744	9.011	9.181

## Data Availability

Data are contained within the article.

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
