# Peer review of "Image Processing Techniques for Improving Quality of 3D Profile in Digital Holographic Microscopy Using Deep Learning Algorithm"

_sensors, 2024, doi:10.3390/s24061950_

Round 1

Reviewer 1 Report

Comments and Suggestions for Authors

Filtering of noisy elevation pixel map by high variance averaging algorithm is considered. The performance of the algorithm is bolstered by applying machine learning approach, which draws better prediction in a specific setup. Some theoretical insights are outlined, experimental results are presented. The language and structure are good.The paper can be published.

Author Response

Thank you for your kind review.

Reviewer 2 Report

Comments and Suggestions for Authors

1. I suggest the authors clarify the terminology used.

Thus, on lines 77 – 80 a typical expression for holography is written for the complex amplitudes of the reference and object waves. Immediately after this we talk about the “subject image” and the “reference image” (lines 81, 90, 92 and further in the text of the article). However, expression (4) is again a standard expression for the interference pattern between the reference and object waves. I suggest that the authors describe their reasoning more clearly.

2. It is proposed at the beginning or at the end of the presentation to describe the developed method in its entirety, in steps and operations, since with a detailed description of each stage of the method, understanding of it as a whole is lost.

3. Apparently, to separate the zero order and side orders in the Fourier spectrum, the mirror of the Mach-Zehnder interferometer is tilted. But this is not discussed in the text, the necessary and permissible angle is not assessed.

4. And a few specific comments.

4.1. The article does not discuss how similar the model chosen for training should be to the object when generating the noise spectrum. Has stability been studied?

4.2. In Fig. 10, the graphics are incorrect regarding the transformation of a parallel beam into a parallel beam by a microlens.

4.3. In Fig. 11 the difference is not noticeable when using Gaussian filtering and HiVA filtering

4.4. In Figures 13 and 14 it is not entirely clear what the straight lines mean. Apparently these are averages, but this needs to be explained.

Comments on the Quality of English Language

Minor editing of English language required

Author Response

Thank you for your kind review.

Reviewer 3 Report

Comments and Suggestions for Authors

General Impression:

This is an interesting paper that seeks to expands the applicability of HiVA algorithm by integrate deep learning method for digital holographic microscopy. This method broaden the scope of denoising while maintaining the denoising efficacy of the HiVA algorithm. However, some corrections need to be revised before publication. There are two points worth discussing and explaining: the drawback of the HIVA algorithm discussed in the article is that it cannot handle holograms with sidebands and DC that are too close in the spectrum; And the accuracy of the model obtained by training the network with the denoised data using the HIVA algorithm. In addition, there is no specific description of the network training process in the paper.

 Specific Comments:

1.     As mentioned before, the authors state that “HiVA can be used for filtering and denoising only when there is sufficient distance between the DC spectrum in the hologram and the sidebands on both sides” in chapter 2.5, and “non-noise data (representing the target data after denoising using HiVA) is utilized for loss function computation”, whether the results of the trained model are accurate when the distance between the DC spectrum in the hologram and the sidebands on both sides is not sufficient.

2.     What is the deep network framework used in this manuscript? What are the specific parameter settings? And what are the parameters and sources of pre-training data? 

3.     What means of φ in Equation (4)? What means of V in Equation (18)? In line 202 V is the variance but in line 203 V is the number of phase reconstruction data.

4.     Figure 8(c) mentioned in line 269 does not exist. 

5.     What does the data number on the horizontal coordinates in Figures 13 and 14 mean?

Comments on the Quality of English Language

None

Author Response

Thank you for your kind review.

Reviewer 4 Report

Comments and Suggestions for Authors

The paper proposed HIVA algorithms to filter DHM noises. Overall, the paper is an interesting paper. The results make sense. Comments are in below.

(1) Overall, the math is not well polished. Please define every symbol and their meaning.

(2) It's hard to follow the diffusion algorithm and its purpose. Please clarify.

(3) How HIVA uses diffusion is also not clear. 

(4) The comparison between different methods are fair is still questionable. Please compare them in more similar standard so we can get an idea for the comparison.

(5) Its confusion to use a chart to present PSNR. Its difficult to interpret. Tables maybe better. 

Comments on the Quality of English Language

Overall English is fine. Minor proofread is needed. 

Author Response

Thank you for your kind review.

Round 2

Reviewer 3 Report

Comments and Suggestions for Authors

Thanks for the author's response. I believe that this version of the manuscript can be published.

Comments on the Quality of English Language

None.

Reviewer 4 Report

Comments and Suggestions for Authors

Authors have revised the paper based on my comments. Therefore, I recommend for acceptance.